# CRONOS: CONTINUOUS TIME RECONSTRUCTION FOR 4D MEDICAL LONGITUDINAL SERIES

**Nico Albert Disch**[1,2,3] , **Saikat Roy**[1,3] , **Constantin Ulrich**[1,5] , **Yannick Kirchhoff**[1,2,3]

**Maximilian Rokuss**[1,3] , **Robin Peretzke**[1,5] , **David Zimmerer**[1,7] , **Klaus Maier-Hein**[1,2,4,6,8]

[1] Division of Medical Image Computing, German Cancer Research Center, Heidelberg, Germany
[2] HIDSS4Health - Helmholtz Information and Data Science School for Health,
Karlsruhe/Heidelberg, Germany
[3] Faculty of Mathematics and Computer Science, University of Heidelberg, Heidelberg, Germany
[4] Pattern Analysis and Learning Group, Department of Radiation Oncology,
Heidelberg University Hospital, Heidelberg, Germany
[5] Medical Faculty Heidelberg, University of Heidelberg, Heidelberg, Germany
[6] Pattern Analysis and Learning Group, Department of Radiation Oncology
Heidelberg University Hospital, [7] Helmholtz Imaging, DKFZ, Heidelberg, Germany
[8] Mohamed Bin Zayed University of Artificial Intelligence, Abu Dhabi, UAE
{nico.disch}@dkfz.de

## ABSTRACT

Forecasting how 3D medical scans evolve over time is important for disease progression, treatment planning, and developmental assessment. Yet existing models either rely on a single prior scan, fixed grid times, or target global labels, which limits voxel-level forecasting under irregular sampling. We present CRONOS, a unified framework for many-to-one prediction from multiple past scans that supports both discrete (grid-based) and continuous (real-valued) timestamps in one model, to the best of our knowledge the first to achieve continuous sequence-to-image forecasting for 3D medical data. CRONOS learns a spatio-temporal velocity field that transports context volumes toward a target volume at an arbitrary time, while operating directly in 3D voxel space. Across three public datasets spanning Cine-MRI, perfusion CT, and longitudinal MRI, CRONOS outperforms other baselines, while remaining computationally competitive. We will release code and evaluation protocols to enable reproducible, multi-dataset benchmarking of multi-context, continuous-time forecasting. Code will be made available under https://github.com/MIC-DKFZ/Longitudinal4DMed

## 1 INTRODUCTION

Longitudinal medical imaging is central to monitoring disease progression, assessing treatment response, and modeling anatomical development across time (Suter et al., 2022; Rivail et al., 2019; Bernard et al., 2018). Some modalities are inherently spatio-temporal, such as ultrasound (US), cine-MRI, videos, or perfusion Computer Tomography (CT). Beyond these, repeated clinical acquisitions form temporal sequences that may span over months or years and are used for clinical decision making. In ophthalmology, for instance, longitudinal OCT volumes are central to monitoring progression of age-related macular degeneration and predicting treatment response (Rivail et al., 2019). Works such as using surgical video streams (Li et al., 2024), which are also increasingly leveraged for diverse tasks, or in (Gomes et al., 2022), where longitudinal US sequences are used, show the overall breadth of spatio-temporal imaging. Beyond individual modalities, there is also a massive and growing amount of video and longitudinal data across clinical contexts (Farhad et al., 2023), including applications such as treatment response prediction in oncology (Suter et al., 2022).

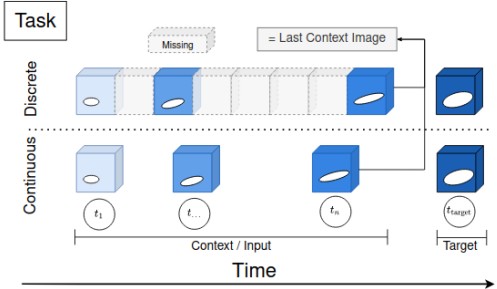

**(a)** Task description.

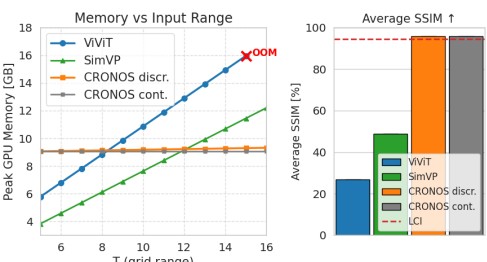

**(b)** Memory Efficiency vs. Performance

Figure 1: **Task and benchmark comparison (a) Task setup** Forecasting a target 3D scan from multiple past volumes in two regimes. *Discrete:* acquisitions lie approximately on a regular grid, but may contain missing frames (dotted boxes). *Continuous:* acquisitions occur at irregular, real-valued timestamps and are used directly without grid alignment. Many-to-one task $(\{I_i\}_{i=1}^T, t_{\text{target}}) \rightarrow I_{\text{target}}$. **(b) Efficiency and performance** Left: GPU memory scaling of single forward pass with sequence length $T$ shows CRONOS to be substantially more memory-efficient than alternatives. Right: Average SSIM across two datasets, where CRONOS outperforms baselines and LCI.

Despite its importance, spatio-temporal learning in medical imaging is centered mostly on single time-point (image-to-image) analysis.

Some approaches rely on global labels e.g. Yoon et al. (2024), while many reduce to image-to-image prediction with a single context scan (Zhang et al., 2025a)). Others introduce task-specific prior or remain tied to one disease (e.g. Puglisi et al. (2025). In particular, Alzheimer's Disease (AD) has attracted a disproportionate share of longitudinal imaging research (Petersen et al., 2010; Martí-Juan et al., 2020; Chen et al., 2025), whereas other domains remain comparatively underexplored.

CRONOS addresses these challenges by introducing a unified spatio-temporal flow framework for medical sequence-to-image prediction that: [1]

| Category | Method | C1 | C2 | C3 | C4 |
|----------|--------|----|----|----|----|
| Med. Gen | BrLP | ✗ | ✓ | ✓ | ✓ |
| | LociDiffCom | ✗ | ✓ | ✓ | ✗ |
| | ImageFlowNet | ✗ | ✓ | ✗ | ✓ |
| Vid. Gen | MCVD | ✓ | ✓ | ✗ | ✗ |
| STL | SimVP | ✓ | ✗ | ✓ | ✗ |
| | ViViT | ✓ | ✗ | ✓ | ✗ |
| | ConvLSTM | ✓ | ✗ | ✓ | ✗ |
| | NODE+LSTM | ✓ | ✗ | ✓ | ✓ |
| Med. STL | CRONOS (ours) | ✓ | ✓ | ✓ | ✓ |

Table 1: **Technical comparison of spatio-temporal prediction methods.** Columns denote Challenges ($C\#$): **C1: Multiple Inputs, C2: High Fidelity, C3: 3D Imaging, C4: Continuous-Time Modeling**. Our proposed CRONOS satisfies all four criteria, whereas existing medical and natural imaging baselines lack one or more. STL stands for spatio-temporal learning.

- **Supports both *discrete* and *continuous* timestamps**, leveraging multiple past scans jointly on **3D** medical imaging data.

- **Avoids disease-specific assumptions**, enabling application to any medical longitudinal task.

- **Consistently outperforms prior approaches**, including standard sequence models and the Last Context Image (LCI) baseline, which is a surprisingly simple and competitive heuristic (NRMSE, PSNR, and SSIM), due to slowly changing medical images.

---

[1]Code will be released at `github.com/anonymous`.

## 2 RELATED WORK

**Medical Imaging**   Prior work in longitudinal medical imaging focuses heavily on one-to-one, or one-to-many video prediction. While approaches like diffusion models (Litrico et al., 2024; Zhu et al., 2024b; Puglisi et al., 2025) and Neural ODEs (Lachinov et al., 2022; Liu et al., 2025) have been applied to medical imaging, these are *image-to-image*, and thus cannot canonically capture multi-input longitudinal evolution. For example, Bai & Hong (2024) propose a continuous-time model, but they predict sequences from single images. In contrast, works that jointly leverage multiple observations show improved prediction accuracy (Fang et al., 2021). The single-context nature makes these aforementioned works not sufficient for our setting. There are also **interpolation-based methods** (Zhu et al., 2024a) which predict intermediate frames between two acquisitions, but this restricts their use to filling missing intervals rather than forecasting. Overall, existing medical approaches are all technically restricted; be it only single-image input, disease specific priors, limited to 2D, or not being able to forecast to arbitrary times as shown in 1.

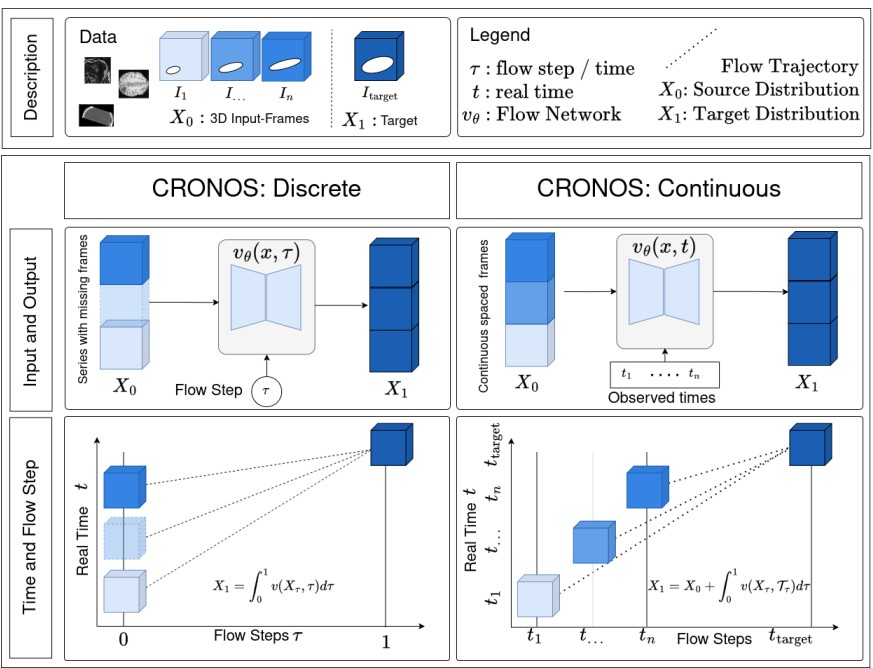

Figure 2: **CRONOS method overview: Left:** Discrete CRONOS treats time implicitly, interpolating between context frames and a fixed target along a normalized flow step $t \in [0, 1]$. **Right:** Continuous CRONOS explicitly conditions on real-valued timestamps $t_i$, allowing each context $I_i$ to transport toward the target via its own interpolation $t_i$. This enables predictions at arbitrary target times while preserving the true temporal geometry.

By contrast, our work focuses on continuous-time modeling across full spatio-temporal sequences without restricting to specific modalities or diseases.

**Natural Imaging and Video Prediction**   Spatio-temporal modeling has been extensively studied in video prediction. Early approaches such as ConvLSTM (SHI et al., 2015) introduced recurrent sequence-to-sequence architectures and remain widely used. Subsequent methods such as SimVP (Gao et al., 2022) replaced recurrence with purely convolutional designs. Transformer-based models like ViViT (Arnab et al., 2021) extended attention mechanisms to the video domain and have become a backbone in many imaging domains. More recent efforts have explored generative modeling, including video diffusion (Voleti et al., 2022; Ye & Bilodeau, 2023; Yan et al., 2021), and continuous-time formulations such as Neural ODEs (Chen et al., 2019), extended to videos in (Park et al., 2021). While these approaches are powerful, they have primarily been developed for dense

2D natural video sequences with large-scale training data. Accordingly, they transfer poorly to 3D medical images with small datasets and sparse sequences , thus motivating our work.

**Flow Matching** Flow Matching (FM) has recently emerged as a generative modeling paradigm (Lipman et al., 2023; 2024), and has been adapted to irregular time series, e.g. in (Zhang et al., 2025b), though only for low-dimensional data rather than full image sequences. Our extension therefore is: while classical FM learn a single flow from (most often) raw noise $X_0 \sim p$ to samples $X_1 \sim 1$ along steps $\tau \in [0, 1]$, we re-cast

$$X_0 = [I_1, \ldots, I_T], \qquad X_1 = \mathcal{I}_{\text{target}} := [I_{\text{target}}, \ldots, I_{\text{target}}], \qquad (1)$$

interpreting $p$ as the context sequence, and $q$ as a broadcast stack of $I_{\text{target}}$ ( defined the stack as $\mathcal{I}_{\text{target}}$, to make dimension explicit). This temporal broadcasting turns FM into sequence-to-image transport: a shared velocity field $v_\theta$ simultaneously moves all $T$ context volumes toward the target, effectively $T$ per-frame transports under shared parameters. We refer to this framework as **Continuous RecOnstructioNs for medical lOngitudinal Series (CRONOS)**.

## 3 METHODS

---

**Algorithm 1** CRONOS Continuous: Training and Inference

---

**Require:** Patients $\mathcal{P}$ and initial network $v_\theta$
 1: **while** training **do**
 2:      Sample $\{[\mathcal{I}, I_{\text{target}}], [t_1, \ldots, t_T, t_{\text{target}}]\} \sim \mathcal{P}(\mathcal{X})$     ▷ pick a random patient
 3:      Sample $\tau \sim \mathcal{U}(0, 1)$     ▷ random flow step
 4:      $\mathcal{I}_{\text{target}} \leftarrow [I_{\text{target}}, \ldots, I_{\text{target}}]$     ▷ repeat target $T$ times
 5:      $\mathcal{T}'_\tau \leftarrow (1 - \tau) [t_1, \ldots, t_n] + \tau\, t_{\text{target}}$     ▷ interpolate timestamps
 6:      $X_\tau \leftarrow (1 - \tau) \mathcal{I} + \tau \mathcal{I}_{\text{target}} + \sigma(\tau)\epsilon$     ▷ linear interpolation
 7:      $\mathcal{L} \leftarrow \|v_\theta(\mathcal{T}'_\tau, X_\tau) - (\mathcal{I}_{\text{target}} - \mathcal{I})\|^2$     ▷ velocity loss
 8:      Update $\theta \leftarrow \text{AdamW}(\nabla_\theta \mathcal{L})$
 9: **return** $v_\theta$
10: **if** inference **then**
11:      Initialize $X_0 \leftarrow \mathcal{I}$
12:      Define integration grid $\{\tau_0 = 0, \ldots, \tau_N = 1\}$ with $N$ steps
13:      $\mathcal{T}'_\tau = (1 - \tau) [t_1, \ldots, t_n] + \tau\, t_{\text{target}}$
14:      $\hat{X}_{0:N} \leftarrow \text{ODEInt}(v_\theta, X_0, \{\mathcal{T}'_0, \ldots, \mathcal{T}'_1\})$     ▷ numerical integration
15:      **return** $\hat{X}_N$

---

### 3.1 PROBLEM SETUP

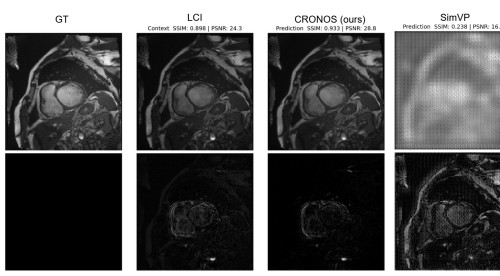

Figure 3: **Qualitative comparison on the ACDC dataset**. Ground truth (GT), Last Context Image (LCI), our method (CRONOS), and SimVP. Upper row: prediction, lower row: residuals.

Let $\mathcal{P} = \left\{ (\{I_i^{(n)}, t_i^{(n)}\}_{i=1}^{T^{(n)}}, t_{\text{target}}^{(n)}, I_{\text{target}}^{(n)}) \right\}_{n=1}^p$ denote a dataset of $p$ patient sequences. Each (patient) sequence consists of a set of $T$ context volumes $\mathcal{I} = \{I_1, \ldots, I_T\}$, with $I_i \in \mathbb{R}^{H \times D \times W}$ (for shorthand $S = H \times D \times W$), acquired at associated timestamps $\{t_1, \ldots, t_T\} \subset \mathbb{R}_+$. We consider two regimes. ***Discrete***: Acquisitions lie on a uniform time grid; some frames may be missing, yielding sparse sequences (e.g., natural video, cine-MRI, perfusion CT). ***Continuous***: Acquisitions occur at irregular, real-valued times that do not easily align to any grid (typical in longitudinal clinical scans). For continuous series, forcing a frame grid either explodes sequence length with empty slots or loses temporal precision. For instance,

daily-resolution for timepoints over several years would yield $T$ in the thousands, yet in practice only a handful of scans are ever acquired. In both discrete and continuous series, $T$ is small relative to natural video.

*Target Task.* Given the set of context images and time $\{(I_i, t_i)\}_{i=1}^T$, as well as a target time $t_{\text{target}}$, we aim to learn

$$f\big(\{I_i, t_i\}_{i=1}^T, \ t_{\text{target}}\big) \mapsto I_{\text{target}}. \tag{2}$$

The discrete setting uses a fixed grid (with optional zero-tensors for missing context volumes); the continuous setting uses the **observed** context only, without padding.

## 3.2 FLOW MATCHING (FM)

Flow Matching Lipman et al. (2023) learns an ordinary differential equation (ODE), linking the equal dimensional distributions $p$ and $q$ via

$$\frac{d}{d\tau}\psi_\tau(x) = u_\tau(\psi_\tau(x)), \qquad X_1 = X_0 + \int_0^1 u_\tau(X_\tau)\, d\tau, \tag{3}$$

with $X_0 \sim p$, $X_1 \sim q$. A convenient coupling is obtained by sampling $X_\tau$ as

$$X_\tau = (1 - \tau)X_0 + \tau X_1 + \sigma(\tau)\epsilon, \tag{4}$$

where $\epsilon \sim \mathcal{N}(0, I)$ denotes random gaussian noise and $\sigma(\tau)$ its intensity, which is sampled around the straight path. The corresponding ground-truth velocity along this path is therefore constant:

$$u_\tau(X_\tau) = \frac{d}{d\tau}X_\tau = X_1 - X_0. \tag{5}$$

Consequently, to approximate the ground truth velocity, we train a neural network $v_\theta(X_\tau, \tau) \in \mathbb{R}^{T \times S}$ using:

$$\mathcal{L}_{\text{CFM}} = \mathbb{E}_{X_0, X_1, \tau}\big\|v_\theta(X_\tau, \tau) - u_\tau(X_\tau)\big\|_2^2. \tag{6}$$

Using $v_\theta$, we can then infer using equation 3 via an approximate ODE solver.

## 3.3 CONTINUOUS AND DISCRETE RECONSTRUCTIONS FOR MEDICAL IMAGE TIME SERIES (CRONOS)

We introduce CRONOS, a spatio-temporal flow model that learns continuous trajectories from longitudinal scans. It comes in two complementary variants: *discrete* and *continuous*.

**Temporal broadcasting for sequence-to-image flows**   To enable flow between a sequence of context images and a single target, we define $X_0 \sim p$ as the stack of context images (with variant-specific handling for continuous vs. discrete), and $X_1 \sim q$ as the target image broadcast to the same shape

$$X_1 = [I_{\text{target}}, \ldots, I_{\text{target}}]. \tag{7}$$

This broadcasting ensures that $X_0$ and $X_1$ share the same dimensionality, allowing us to define a valid flow between them.

**Discrete CRONOS.**   On a regular grid with missing scans, we first *embed* each sequence onto the grid of a resolution g using a binning operator $\mathcal{E}_{\mathbf{g}}^{\text{grid}}$, which assigns each $I_i$ to the closes grid index matching $t_i$ (proper definition in A.1.1). Missing slots are then handled by a last-observed carry-forward operator $\mathcal{F}^{\text{LOCF}}$, which fills empty positions with the most recent available scan. In short, we define

$$X_0 = \big(\underbrace{\mathcal{F}^{\text{LOCF}}}_{\text{fill}} \circ \underbrace{\mathcal{E}_{\mathbf{g}}^{\text{grid}}}_{\text{bin to grid}}\big)\big(\{(I_i, t_i)\}_{i=1}^T\big) = [\hat{I}_1, \ldots, \hat{I}_K]. \tag{8}$$

This pre-processing ensures $X_0$ is well-defined on a uniform grid. Furthermore, LOCF handles spatial missingness: missing frames are zero-initialized and replaced by the most recent observation (Appendix A.1.2). This setup stabilizes optimization and preserves grid order while enabling many-to-one sequence transport within FM. Finally, we train on the linear interpolation $X_\tau = (1-\tau)X_0 + \tau X_1$ using equation 6, where temporal order is captured *implicitly* by the flow step $\tau$ and the frame index. Additionally, we set $\sigma = 0$ during training and inference, ablations on nonzero noise levels are reported in Table 7.

**Continuous CRONOS** Our continuous modeling strategy extends on the discrete case by conditioning on *real-valued timestamps* while evolving along a scalar flow parameter $\tau \in [0, 1]$. We construct spatio-temporal tensors (using mild abuse of notation): Time enters the network only as conditioning on real timestamps. We interpolate the conditioning timestamps along the interpolated time vector $\mathcal{T}_\tau$. $X_0$ is then defined as in equation equation 7, without embedding it to the grid, nor performing LOCF as in discrete CRONOS. We define the shifted time vector as

$$\mathcal{T}_\tau = (1 - \tau)\, \mathbf{t}_{\text{ctx}} + \tau\, \mathbf{t}_{\text{target}}. \tag{9}$$

The formulation in equation 9 lets flow step $\tau$ carry real temporal information, without adding extra complexity. The conditional trajectory is then

$$X_1 = X_0 + \int_0^1 v_\theta\big(X_\tau, \mathcal{T}_\tau\big)\, d\tau, \tag{10}$$

where $v_\theta$ is the predicted velocity field and $\tau$ *is the flow step* (usually called time, we avoid it due to avoiding confusion). Prediction is then done via approximate solution of equation 3, solver details found in C.1. This formulation lets CRONOS model continuous image evolution grounded in actual scan times, supporting interpolation or forecasting without regular sampling or artificial frame filling. It avoids zero-padding, leading to reduced computational burden compared to the discrete variant. Both variants use the same 3D U-Net backbone, further details are provided in Appendix C.3, and the training/inference procedure appears in Algorithm 1.

**Time Encoding.** Flow steps and continuous times are mapped to Fourier embeddings using (Tancik et al., 2020), which were used e.g. in (Rombach et al., 2022): $\gamma(t) = [\sin(2\pi f_k t), \cos(2\pi f_k t)]_{k=1}^K$ using frequencies $f_k$. To preserve dimensional consistency across variable-length input sequences for the continuous setting, we compute the time embedding as

$$\text{Enc}(\boldsymbol{t}) = \frac{1}{T} \sum_{i=1}^T \gamma(t_i). \tag{11}$$

This embedding is then added to each residual layer via FiLM. The loss is then calculated via equation 6, and inference via equation 10.

## 4 DATA AND EXPERIMENTAL DESIGN

### 4.1 DATASETS

**ACDC** (Bernard et al., 2018) is a cardiac MRI dataset capturing different heart phases. The context tensor is reshaped to $[T, H, D, W] = [11, 32, 128, 128]$, and the target is a single image with the same spatial size. We split ACDC into 80 training, 20 validation and 50 test images. This dataset served for method development; ablations were conducted on the validation split.

**ISLES** (Riedel et al., 2024) consists of perfusion CT image time series from stroke patients. From the normalized series, we sample 7 consecutive points, take the last as the target, and randomly mask the remaining context frames. The resulting context tensor has shape $[T, H, D, W] = [7, 16, 128, 128]$. We use a split of 92 training, 23 validation and 34 test images. For both the ACDC and dataset, we randomly mask out time points (see Appendix C.2).

**Lumiere** (Suter et al., 2022) is a longitudinal glioma MRI dataset with 3D scans. Images are reshaped to $[T, H, D, W] = [7, 96, 96, 64]$. Because some patients have few acquisitions, we prepend zeros to standardize pre-processing across cases. The split is 48 training, 12 validation and 14 test images.

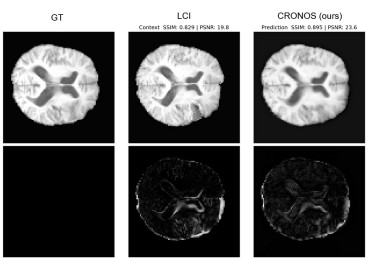

Figure 4: **Qualitative comparison on the LUMIERE dataset**. Ground truth (GT), Last Context Image (LCI), our method (CRONOS), and SimVP baseline. Lumiere is particularly challenging due the very small dataset. highlighting the benefit of explicit continuous-time conditioning under extreme data scarcity.

## 4.2 EXPERIMENTAL SETTINGS

Reproducibility details can be found in Section C.

**Discrete Setting**: As mentioned in the data section, input data has dimension $T$, while some frames may be missing. We apply *both* variants of CRONOS, noting that the continuous version can also operate in this regime with a smaller context window, since missing images *do not need* to be explicitly represented. The lower context window also leads to a lower computational demand. Therefore, the underlying tensors remain uniform, with some time points masked. For validation and testing we ensure that the missingness pattern is fixed across epochs, as otherwise the choice of best checkpoint would be ill-posed (further details in Appendix C.2).

**Continuous Setting**: As an *additional ablation and experiment*, we simulate a continuous setup on ACDC to highlight the gains from explicit timestamp conditioning. While no public dataset provides plenty of continuous acquisition protocols, this sub-sampled variant shows that CRONOS benefits from real-valued time even beyond irregular masking. Specific details of how we subsampled ACDC can be found in C.1. Importantly, both the discrete and continuous formulations remain applicable to discrete grids.

Table 2: **Discrete Time: Quantitative Evaluation on Many-to-One Sequences:** Reported values are mean (standard deviation) over three runs. Metrics include normalized root $MSE$, $NRMSE$, structural similarity index ($SSIM[\%]$) and peak signal-to-noise-ratio $PSNR$. *ViViT OOM on a 40 GB GPU, despite having a smaller batch size and the lowest possible feature size. Standard deviation of LCI omitted for visual clarity. Blue row: only method to beat LCI and our proposed CRONOS. Computational requirements on ACDC in A

| Dataset | Model | NRMSE [$10^{-2}$] ↓ | SSIM [%] ↑ | PSNR [$dB$] ↑ |
|---|---|---|---|---|
| ACDC | LCI | 4.48 | 92.79 | 28.918 |
| | ConvLSTM | 11.20 ± 0.48 | 50.44 ± 1.53 | 19.123 ± 0.312 |
| | SimVP | 9.27 ± 0.29 | 49.08 ± 4.01 | 20.715 ± 0.267 |
| | NODE + LSTM | 11.59 ± 0.18 | 36.41 ± 2.94 | 18.946 ± 0.186 |
| | ViViT | 13.90 ± 2.66 | 17.06 ± 8.60 | 17.252 ± 1.738 |
| | CRONOS discrete | 3.97 ± 1.23 | **94.51 ± 0.79** | **30.510 ± 1.560** |
| | CRONOS cont. | **3.74 ± 0.21** | 94.34 ± 0.45 | 29.750 ± 0.528 |
| ISLES | LCI | 5.25 | 96.29 | 29.002 |
| | ConvLSTM | 19.31 ± 0.18 | 39.92 ± 0.66 | 17.644 ± 0.014 |
| | SimVP | 13.06 ± 0.19 | 48.82 ± 1.60 | 20.799 ± 0.112 |
| | ViViT | 16.54 ± 0.30 | 36.76 ± 1.49 | 18.671 ± 0.134 |
| | NODE + LSTM | 15.10 ± 0.87 | 40.55 ± 7.15 | 19.481 ± 0.515 |
| | CRONOS discrete | 4.50 ± 0.76 | **97.33 ± 0.93** | 30.542 ± 1.540 |
| | CRONOS cont. | **4.38 ± 0.48** | 97.31 ± 0.38 | **30.809 ± 1.099** |
| Lumiere | LCI | 8.38 | 88.35 | 21.631 |
| | ConvLSTM | 34.79 ± 0.67 | 9.21 ± 2.81 | 9.217 ± 0.171 |
| | SimVP | 71.03 ± 0.89 | -1.92 ± 0.51 | 2.989 ± 0.109 |
| | ViViT* | OOM | OOM | OOM |
| | NODE+LSTM | 13.07 ± 1.03 | 48.66 ± 2.26 | 17.742 ± 0.659 |
| | CRONOS discrete | 7.92 ± 0.92 | **91.43 ± 1.84** | 22.427 ± 0.969 |
| | CRONOS cont. | **7.55 ± 0.86** | 89.32 ± 1.83 | **22.551 ± 0.979** |

**Baselines** We compare CRONOS against established spatio-temporal learning methods. As a clinically motivated heuristic, the Last Context Image baseline (LCI) simply reuses the last available image and serves as a lower bound. Among sequence models, we include ConvLSTM (SHI et al., 2015), SimVP (Gao et al., 2022), and ViViT (Arnab et al., 2021) as representative recurrent, convolutional, and transformer backbones. For continuous-time sequence modeling, we further evaluate an ODE-LSTM (Lechner & Hasani, 2020) baseline. For the flow matching library we use Tong et al. (2024b;a); Tong (2025). Together, these methods provide a spectrum of spatio-temporal architectures against which we benchmark CRONOS. Computational requirements are described in detail in the appendix.

**Continuous vs. Discrete.** We report results in two regimes: an *discrete* setting, which allows direct comparison to existing spatio-temporal baselines, and a *continuous* setting on ACDC, designed as an ablation to test the benefit of explicit timestamp conditioning.

## 5 RESULTS AND DISCUSSION

### 5.1 TOWARDS UNIFIED BENCHMARKING FOR MEDICAL 3D SEQUENCE-TO-IMAGE FORECASTING

We are among the first to propose an experimental setup for the sequence-to-image task, evaluating CRONOS under two complementary regimes. The first uses *discrete* input sequences, where some context images are missing but acquisitions lie on a regular grid. This setting enables comparison against established spatio-temporal baselines. The second uses ACDC with resampled acquisitions to mimic *continuous* input, allowing us to assess the benefit of explicit timestamp conditioning. For completeness, we include an image-to-image (*not sequence-to-image*) diffusion baseline on ACDC (details in B.5) This required a two-stage training setup, first pretraining an autoencoder and then training the diffusion module for 1000 denoising steps, which already made the approach far more computationally demanding than all other baselines. Iterative denoising leads to an order-of-magnitude longer inference time for a single image-to-image step and several orders of magnitude higher training cost, while not surpassing the simple LCI heuristic. [2]

### 5.2 CRONOS IS STATE-OF-THE-ART FOR SPATIO-TEMPORAL 3D MEDICAL IMAGE FORECASTING

Table 2 reports the quantitative results across all three datasets. We observe that both variants of CRONOS **substantially outperform** the competing spatio-temporal baselines, as well as LCI. We also note that individually, CRONOS is better than LCI on each individual validation run. On LUMIERE, which is characterized by very sparse and heterogeneous tumor trajectories, it is surprising that CRONOS is even able to outperform LCI. These results demonstrate that CRONOS is effective across different temporal regimes: the discrete formulation already yields strong performance, while the continuous formulation provides further gains when

| Method | SSIM ↑ | PSNR ↑ | NRMSE ↓ |
|---|---|---|---|
| LCI | 93.27 | 29.77 | 0.0349 |
| NODE + LSTM | 57.50 | 22.87 | 0.0728 |
| CRONOS discr. | 93.27 | 29.77 | 0.0348 |
| CRONOS cont. | 93.86 | 30.09 | 0.0330 |

Table 3: **Continuous ACDC**, where discrete CRONOS lacks explicit timestamp conditioning, and therefore fails to outperform LCI. Additional experiments in B and in Table 9

timestamps are informative. CRONOS runs **within the same computational budget** during inference (see Figure 1b) and in similar orders of magnitude (VRAM and wall-clock time) during training as natural imaging baselines (see 8). Further ablations are provided in A, confirming that CRONOS is stable across variations in *feature size, training noise, and integration settings*. While small differences appear, they are not substantial, indicating that our network is *highly robust* to hyperparameter choices.

### 5.3 CRONOS ENABLES EFFICIENT FLOW-BASED CONTINUOUS MEDICAL MODELING

Table 3 demonstrates that *incorporating explicit time embeddings improves forecasting* quality when scans occur at irregular intervals. This shows that the continuous formulation of CRONOS is not only feasible but also beneficial in realistic clinical settings, where images are often irregularly sampled. In fact, if we fully remove the timestamp information entirely, performance differences increase significantly, and the continuous variant clearly outperforms the discrete one 9. Together, these results highlight that modeling real-valued timestamps can provide a measurable advantage over treating sequences as grid-aligned. However, in Table 2, we see that using the discrete variant

---

[2]On our setup, a naive auto-regressive image-to-image *latent-diffusion* pipeline applied across 11 context times per subject requires ∼5–6 hours *per validation step*; see Appendix A for details.

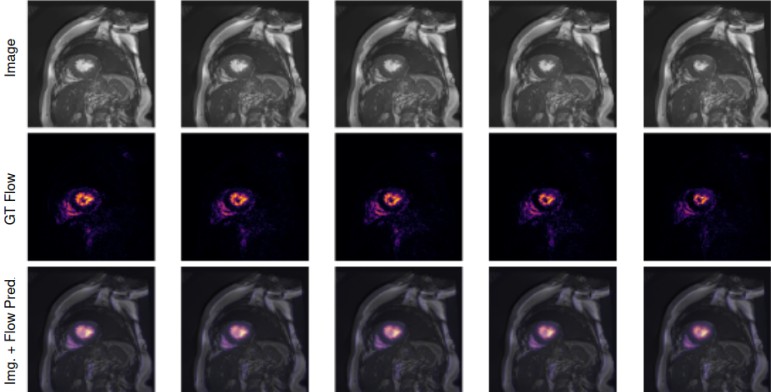

Figure 5: **Network Flows**: Top: input images at the first five timestamps. Middle: ground-truth voxel-wise differences ($|I_i - I_{target}|$). Bottom: predicted velocity fields $v_\theta(X_0, 0)$, overlaid on the corresponding inputs. The highlighted regions coincide with the areas of the largest temporal changes (primarily the ventricular cavities and myocardial boundaries).

remains highly competitive. Although any irregular series can in principle be quantized to a grid via $\mathcal{E}_{\mathbf{g}}^{\text{grid}}$, doing so without loss requires increasingly fine grids. This becomes computationally inefficient, whereas the continuous variant scales with the number of context images and *not* with the grid range $K \cdot \Delta$ (see equation 12). This is reflected in Table 8 and Figure 1b, where continuous CRONOS is both more memory-efficient and faster to train than the discrete formulation. It also highlights a broader limitation of the field: the scarcity of diverse spatio-temporal datasets in which real timing information is critical.

## 5.4 CRONOS PRODUCES SHARPER RECONSTRUCTIONS WIT LOWER RESIDUALS

Figures 3, 6 and 4 highlight qualitative comparisons, as well as dataset examples. The LCI baseline often appears visually close to the target, largely because many longitudinal scans exhibit only subtle changes. However, e.g. SimVP tends to introduce artifacts and blur anatomical details. In contrast, CRONOS yields sharper reconstructions and consistently lower residuals compared to LCI, highlighting its ability to capture fine-grained temporal progression.

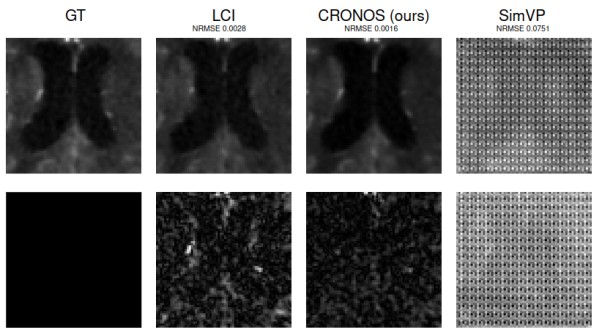

Figure 6: **Zoomed-in qualitative comparison on the ISLES dataset**. Ground truth (GT), Last Context Image (LCI), our method (CRONOS), and SimVP baseline. Shown here for visibility is a zoomed in patch of the qualitative results of the ISLES dataset.

## 5.5 FUTURE WORK: UNLOCKING GENERAL SPATIO-TEMPORAL MEDICAL FORECASTING

While voxel-wise fidelity metrics such as NRMSE, PSNR, and SSIM remain the community standard, they do not fully capture clinically relevant trajectory modeling. As highlighted in recent efforts on image analysis validation Maier-Hein et al. (2024), such metrics may not always align with actual domain interest. Developing metrics for spatio-temporal forecasting is therefore an important future direction. In parallel, the scarcity of longitudinal and spatio-temporal datasets (beyond the ones we used in this study), poses a broader challenge for robust evaluation. Encouragingly, our results on LUMIERE suggest that progress is

possible even under severe data limitations, and we hope to motivate further work on curating larger and more diverse publicly available cohorts. Finally, the absence of large-scale foundation models for medical imaging, particularly in the spatio-temporal domain, remains a major bottleneck. We view our work as a keystone contribution: establishing a unified flow-based framework for continuous spatio-temporal medical volumetric forecasting that can *both benefit from, and motivate*, future developments in medical imaging.

## 6 CONCLUSION

In this work, we presented CRONOS (Continuous RecOnstructioNs for medical lOngitudinal Series), a unified spatio-temporal framework that forecasts 3D medical volumes at arbitrary target times by combining multiple context scans with explicit real-valued time conditioning. Unlike single-image or time-agnostic methods, CRONOS handles both grid-aligned and continuous timestamps within one architecture, and makes no disease-specific assumptions. It is among the first methods to demonstrate continuous sequence-to-image forecasting for 4D medical data. Across three publicly available datasets (Cine-MRI, perfusion CT, longitudinal MRI), it outperforms baselines-including the strong Last Context Image (LCI)-and remains robust under hyperparameter changes while remaining computationally competitive. By resolving the aforementioned limitations, our method enables clinically specific studies and advances patient-level forecasting for personalized precision medicine.

## BROADER IMPACT

Longitudinal modeling of medical images has the potential to improve patient care by enabling earlier detection of disease progression, monitoring of treatment response, and improved personalization of therapy. By explicitly modeling continuous temporal evolution, our approach could support clinicians in making more informed decisions. However, there are also risks: mispredictions may lead to incorrect clinical conclusions if models are deployed without careful validation and without a human in the loop. Biases in training data (e.g., underrepresentation of certain populations or imaging modalities) may propagate to predictions, raising concerns about fairness and generalizability, which is a common problem in medical imaging. We emphasize that our method is a research contribution intended to advance especially technical methodology. Clinical deployment would require extensive validation, regulatory approval, and integration into existing workflows. We believe that by releasing code and benchmarks, this work will support the community in building transparent, reproducible, and safe spatio-temporal models for healthcare. But by proposing this method, we hope to support a general-purpose foundation for medical spatio-temporal and longitudinal modeling, which could massively propel this area forward.

## ACKNOWLEDGMENTS

The present contribution is supported by the Helmholtz Association under the joint research school "HIDSS4Health – Helmholtz Information and Data Science School for Health. N.A.D. and Y.K. were funded by HIDSS4Health, D.Z. was funded by HELMHOLTZ IMAGING, a platform of the Helmholtz Information & Data Science Incubator, C.U. was funded by the Helmholtz Foundation Model Initiative (HFMI) through the pilot project THRP (The Human Radiome Project), M.R. was funded via the Phd Program DKFZ, R.P. was funded by the Deutsche Forschungsgemeinschaft (DFG, German Research Foundation) – 536508040.

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

# A  APPENDIX

## A.1  DISCRETE CRONOS DETAILS

### A.1.1  GRID EMBEDDING

Assume we want to embed data on a grid with spacing $\Delta > 0$, and a maximum size of $K \in \mathbb{N}_+$, where

$$g_k = g_1 + (k-1)\Delta, \quad k \in \{1, \ldots, K\}. \tag{12}$$

In total, we define the grid as $\mathbf{g} = [g_1, \ldots, g_k]$. We define the grid quantizer as

$$q(t_i) = \mathrm{clip}\left(1 + \left\lfloor \frac{t_i - g_1}{\Delta} + \tfrac{1}{2} \right\rfloor, 1, K\right), \tag{13}$$

where $\mathrm{clip}(a, b, c) := \min(\max(a, b), c)$. I.e. we clip the value $t$ to the closest grid point $g_k$. We define the Kronecker delta for indices

$$\delta_{k, q(t_i)} = \begin{cases} 1, & q(t_i) = k \\ 0, & \text{else.} \end{cases} \tag{14}$$

Since for some grid sizes $\Delta$, there can be too many items, we define the occupancy:

$$m_k = \min\left(1, \sum_{i=1}^{T} \delta_{k, q(t_i)}\right) \in \{0, 1\}. \tag{15}$$

We use the last available index for filling:

$$i^*(k) = \mathrm{argmax}_i(\delta_{k, q(t_i)} t_i), \tag{16}$$

i.e. the index which is *latest* while still falling into the index $k$. Finally, our embedded images are

$$\tilde{I}_k = \begin{cases} I_{i^*(k)}, & m_k = 1 \\ 0, & \text{else.} \end{cases} \tag{17}$$

Lastly we define the grid embedding as

$$\mathcal{E}_{\mathbf{g}}^{\mathrm{grid}}\big(\{(I_i, t_i)\}_{i=1}^{T}\big) = [\tilde{I}_1, \ldots, \tilde{I}_K]. \tag{18}$$

### A.1.2  LAST OBSERVED CARRY FORWARD.

Longitudinal imaging sequences are often incomplete, with missing acquisitions at certain timestamps. To ensure consistent tensor inputs, we adopt the last-observed carry-forward procedure: missing frames are initialized as zero images and then replaced by the nearest available context scan in time. We iteratively define the

$$\hat{I}_1 = \tilde{I}_{k_0}, \quad \hat{I}_k = m_k \tilde{I}_k + (1 - m_k)\hat{I}_{k-1} \quad k \in \{2, \ldots, K\}, \tag{19}$$

where $k_0$ defines the first observed image in the grid

$$k_0 = \min\{k \in \{1, \ldots, K\}\} | m_k = 1\}. \tag{20}$$

We define shorthand:

$$\mathcal{F}^{\mathrm{LOCF}}\left([\tilde{I}_1, \ldots, \tilde{I}_K]\right) = [\hat{I}_1, \ldots, \hat{I}_K]. \tag{21}$$

This guarantees that the model always observes a full sequence $\{I_1, \ldots, I_T\}$ while still preserving the true irregularity of acquisition. Empirically, sparsity filling stabilizes training and improves performance compared to leaving missing slots empty, since it prevents the network from confusing acquisition gaps with valid image content. This can be seen in the ablations, see Table 4.

### A.1.3  PRACTICAL IMPLEMENTATION

For datasets which are regular, this grid embedding is simply achieved by randomly masking some entries. The occupancy operator also highlights one key trade-off issue: if $\Delta$ is too large, we possibly overwrite some images. If $\Delta$ is too small, our context window is also very small if we keep within a budget of $K$ tensor size.

---

**Algorithm 2** CRONOS: Discrete training and Inference

---

**Require:** Patients $\mathcal{P} = \{[\mathcal{I}_1, I_{\text{target},1}], \ldots, [\mathcal{I}_p, I_{\text{target},p}]\}$ and initial network $v_\theta$

 1: **while** training **do**
 2: $\quad$ $[\mathcal{I}, I_{\text{target}}] \sim \mathcal{P}(\mathcal{X})$ $\hfill \triangleright$ pick a random patient
 3: $\quad$ $\tau \sim \mathcal{U}(0,1)$ $\hfill \triangleright$ pick a random flow step
 4: $\quad$ $\mathcal{I}_{\text{target}} \leftarrow [I_{\text{target}}, \ldots, I_{\text{target}}]$ $\hfill \triangleright$ Extend the dimension of $I_{\text{target}}$ $T$ times
 5: $\quad$ $\mathcal{I}' \leftarrow \text{LOCF}(\mathcal{I})$ $\hfill \triangleright$ Fill empty images
 6: $\quad$ $X_\tau \leftarrow (1-\tau)\,\mathcal{I}' + \tau\,\mathcal{I}_{\text{target}} + \sigma$ $\hfill \triangleright$ Calculate the linear interpolation
 7: $\quad$ $\mathcal{L}_{\text{FM}} \leftarrow \big\| v_\theta\big(\tau,\, X_\tau\big) - (\mathcal{I}_{\text{target}} - \mathcal{I}')\big\|^2$ $\hfill \triangleright$ Calculate the velocity loss
 8: $\quad$ Update $\theta \leftarrow \text{AdamW}(\nabla_\theta \mathcal{L}_{\text{FM}})$
 9: **return** $v_\theta$
10: **if** inference **then**
11: $\quad$ Initialize $X_0 \leftarrow \mathcal{I}'$
12: $\quad$ Define integration grid $\{\tau_0 = 0, \ldots, \tau_N = 1\}$ with $n$ steps
13: $\quad$ $\hat{X}_{0:N} \leftarrow \text{ODEInt}\big(v_\theta, X_0,\, \{\tau_0, \ldots, \tau_N\}\big)$ $\hfill \triangleright$ numerically integrate the network
14: $\quad$ **return** $\hat{X}_N$

---

Table 4: **Ablation Results for discrete CRONOS on ACDC:** This table compares TFM under different design changes, showing the performance under each scenario. The ablations were done on an ACDC validation set. We evaluate the effect of using a more lightweight version of the U-Net which does not use attention('No Att'). Instead, $\tau$ and image embeddings are merged via concatenation in the bottleneck. We also compare aggregating via the mean and the last image, but these results are only for inference. Training is still done the same way. Third, we compare LOCF with the alternative of using the image sequences $\mathcal{I}$ as they are given, which notably reduces performance. *Limiting the model to only see LCI during training and perform FM on this is unstable, which highlights the importance of temporal context.

| Change | NRMSE ($\downarrow$) | SSIM ($\uparrow$) | PSNR ($\uparrow$) |
|---|---|---|---|
| Att U-Net & Mean | 0.0261 | 96.04 | 32.30 |
| No Att: Mean | 0.0270 | 95.77 | 31.88 |
| No Att: Last | 0.0271 | 95.77 | 31.87 |
| No LOCF + Masking | 0.0380 | 95.55 | 31.20 |
| No LOCF | 0.0444 | 90.92 | 27.30 |
| LCI | 0.0380 | 93.50 | 29.49 |

## B  FURTHER ABLATIONS

### B.1  CRITICAL ABLATIONS

In Table 4 we see the different design choices of the discrete CRONOS. Notably, LOCF or masking (which is training only on the given frames), achieves the biggest boost versus the baseline. We note that the masked training takes longer to converge, but yields a similar, albeit not the same performance.

### B.2  ZERO-SHOT ROBUSTNESS TO MASKING ORDER

In Figure 7, we evaluate how prediction quality changes when masking context frames from early to late. As expected, increasing the temporal gap between the last available frame and the target leads to a monotonic decrease in SSIM. Interestingly, when the final frame remains present and only earlier frames are removed, the curve becomes U-shaped: performance peaks when roughly five context frames are masked. This aligns with the masking distribution used during training, where such levels of sparsity were most common. This also suggests that the model is optimized for the typical irregular pattern seen during training, and the fully unmasked case is not the easiest *in the zero-shot* evaluation.

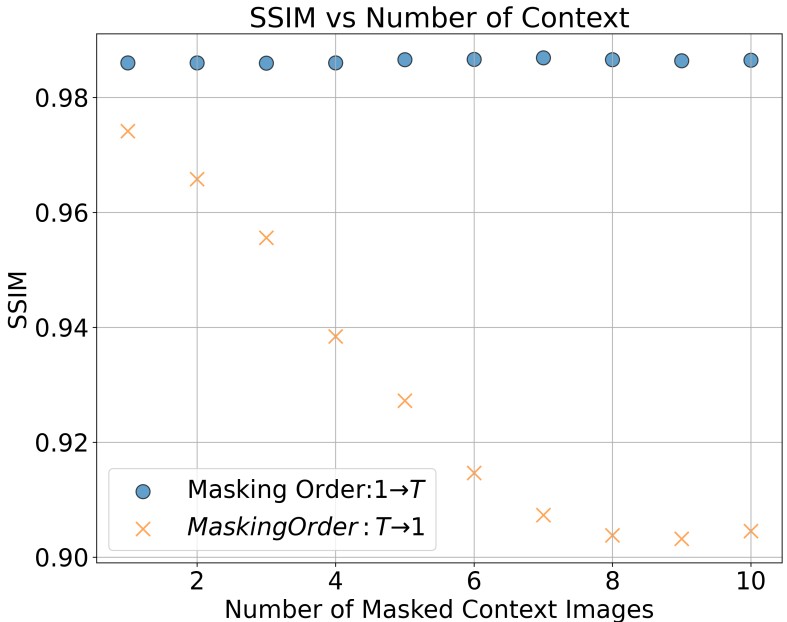

Figure 7: **Zero-shot masking of early vs. late context frames on ACDC.** We compare CRONOS's prediction quality when masking context frames at inference time. In the early masking protocol $(1 \rightarrow T)$, we remove the earliest context frames first, in the late masking protocol $(T \rightarrow 1)$, we remove the frames closest to the target. The x-axis denotes the total number of masked frames. Masking frames near the target causes a clear, monotonic drop in SSIM, confirming that CRONOS relies on temporally proximal information. Masking from earlier frames leads only to a slight degradation, and shows a weak U-shaped behavior at larger mask counts.

### B.3  HYPERPARAMETER ABLATIONS

We did ablation studies on one ACDC validation subset.

Table 5: Ablation on feature size (FS) for the proposed method, compared to the last context image (LCI) baseline. Values in parentheses indicate the difference relative to LCI. Larger FS values generally improve NRMSE, SSIM, and PSNR, with FS = 32.0 achieving the best overall performance.

| Feature Size | NRMSE ($\downarrow$) | SSIM ($\uparrow$) | PSNR ($\uparrow$) | Max Mem[$GB$] |
|---|---|---|---|---|
| LCI | 0.038 | 93.50 | 29.49 | - |
| FS=8.0 | 0.027 (-0.012) | 95.94 (+2.38) | 32.07 (+2.74) | 2.70 |
| FS=16.0 | 0.026 (-0.013) | 95.99 (+2.43) | 32.17 (+2.83) | 3.27 |
| FS=32.0 | 0.026 (-0.013) | **96.06 (+2.49)** | 32.24 (+2.90) | 7.70 |
| FS=64.0 | **0.026 (-0.014)** | 95.97 (+2.41) | **32.35 (+3.01)** | 11.21 |

Table 6: **Evaluating Metrics vs. Number of Function Evaluations:** We evaluate how the number of function evaluations (NFEs) affects $SSIM$ performance on one ACDC validation set. $SSIM$ increases with more evaluations and peaks at $5 - 10$ NFEs, after which it plateaus. However, the improvement becomes marginal beyond just 5 NFEs. As trade-off we use 10 NFE throughout.

| NFE | NRMSE ($\downarrow$) | SSIM ($\uparrow$) | PSNR ($\uparrow$) |
|---|---|---|---|
| LCI | 0.038 | 93.50 | 29.49 |
| NFE=1.0 | 0.030 (-0.009) | 95.43 (+1.87) | 31.47 (+2.13) |
| NFE=5.0 | **0.025 (-0.014)** | **96.18 (+2.62)** | **32.47 (+3.13)** |
| NFE=10.0 | 0.026 (-0.013) | 96.06 (+2.49) | 32.24 (+2.90) |
| NFE=100.0 | 0.026 (-0.013) | 96.04 (+2.48) | 32.30 (+2.97) |
| NFE=200.0 | 0.025 (-0.014) | 96.18 (+2.62) | 32.47 (+3.13) |

Table 7: **Training noise amount** validation performance ablation on training noise (TN) levels for the proposed method, compared to the last context image (LCI) baseline. Values in parentheses indicate the difference relative to LCI. Moderate TN levels (0.0–0.05) consistently improve NRMSE, SSIM, and PSNR, with TN = 0.0 yielding the best overall performance.

| Training Noise | NRMSE ($\downarrow$) | SSIM ($\uparrow$) | PSNR ($\uparrow$) |
|---|---|---|---|
| LCI | 0.039 | 93.53 | 29.42 |
| TN=0.0 | 0.026 (-0.013) | 96.06 (+2.49) | 32.24 (+2.90) |
| TN=0.01 | **0.026 (-0.014)** | **96.12 (+2.56)** | 32.32 (+2.98) |
| TN=0.025 | 0.026 (-0.013) | 96.06 (+2.49) | **32.34 (+3.00)** |
| TN=0.05 | 0.026 (-0.013) | 96.04 (+2.47) | 32.26 (+2.92) |
| TN=0.1 | 0.026 (-0.013) | 96.06 (+2.49) | 32.24 (+2.90) |
| TN=0.3 | 0.026 (-0.013) | 95.78 (+2.21) | 32.26 (+2.92) |

In Table 7, we see the different effects of training noise on the performance. While we find that $0.01$ seems to yield the best performance, we kept the noise at $0$, in order to compare to other baselines, which were not trained in a noisy fashion.

## B.4 RUNTIME AND MEMORY USAGE

To provide a transparent comparison of computational efficiency, we report wall-clock training time and peak GPU memory for a single run of each baseline. All experiments were run on the same hardware with batch size 4. Table 8 shows that CRONOS achieves competitive runtime and memory usage compared to other spatio-temporal baselines, while diffusion-based approaches remain significantly more expensive (see main text).

## B.5 *Image to image comparison*

In Figure 8, we include a diffusion baseline adapted from Puglisi et al. (2025). Diffusion models are widely used for generative modeling, but their iterative denoising leads to orders of magnitude

| Method | Runtime $[10^3\text{s}]$ | Max Memory [GB] |
|---|---|---|
| ViViT | 16 | 9.0 |
| NODE+LSTM | 60 | 6.6 |
| SimVP | 20 | 5.2 |
| ConvLSTM | 14 | 3.5 |
| CRONOS irregular | 18 | 7.7 |
| CRONOS continuous | 12 | 7.0 |

Table 8: Wall-clock runtime in thousands of seconds and maximum memory usage for a single run.

Figure 8: **Image to Image - ACDC.** Efficiency–performance trade-off on ACDC: PSNR (↑) vs. training time (GPU hours, log scale). The red dashed line marks the LCI baseline; red/blue shading indicates better/worse than baseline. Bubble size encodes VRAM needs. *Ours CRONOS (tiny)* achieves above-baseline PSNR with roughly two orders-of-magnitude less training than Latent Diffusion AE, which remains below the baseline despite substantially higher cost.

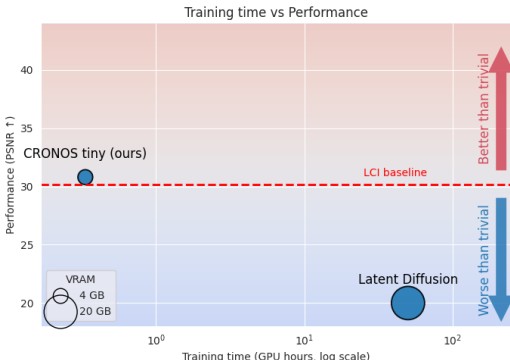

longer inference times and, in our setting, they do not outperform the simple LCI heuristic. *Most importantly*, to the best of our knowledge, **no prior** 3D continuous time diffusion approach in medical imaging **supports forecasting from multiple input volumes**. We therefore present diffusion here as a motivating image-to-image reference, but focus the remainder of our study on spatio-temporal learning (STL) methods, which are both more computationally feasible and explicitly designed for multi-context forecasting. So, the results from 8, the inference would take roughly 35 minutes, and auto-regressive diffusion would take $5-6\,h$ in total per sampling step

update fig- ure

### B.6 CONTINUOUS ABLATION

#### B.6.1 MAIN RESULTS ABLATION

For the headline comparison we further restrict to a *single* context frame (image-to-image forecasting), which is markedly harder for the discrete CRONOS than multi-context prediction, as the last-observed carry-forward is more challenging. We keep the same training objective and evaluation protocol; see Tab. 3 for metrics under this stricter setting. We observe that the continuous CRONOS is better than LCI, while the discrete one is not. Even stronger prediction can be seen in 9.

#### B.6.2 ADDITIONAL ABLATIONS FOR USE OF CONTINUOUS TIME

To isolate the effect of explicit time, we construct a continuous setting where only the continuous variant has access to time information. We use a 4-frame grid, where we uniformly sample 2 images. From the missing future frames, we predict a single future frame. Here, the discrete model receives *no timestamps*. The continuous model receives the two context times and the target time during inference. All else is being equal (architecture, training regime, etc.). Results in Tab. 9 show that explicit timestamps are necessary in this protocol.

Table 9: **Continuous conditioning improves sequence-to-image forecasting.** Comparison of the LCI baseline with CRONOS under discrete (implicit time) and continuous (explicit timestamps) settings. Metrics are reported as SSIM↑, PSNR↑, NRMSE↓. The continuous variant attains the best scores across all metrics, indicating that real-valued time conditioning is effectively exploited.

| Method | SSIM ↑ | PSNR ↑ | NRMSE ↓ |
|---|---|---|---|
| LCI (baseline) | 0.94997 | 31.727 | 0.03038 |
| CRONOS discrete | 0.96671 | 33.534 | 0.02211 |
| CRONOS continuous | 0.97814 | 36.007 | 0.01670 |

## C    REPRODUCIBILITY

We will describe the experiments to further facilitate reproducibility.

### C.1    EXPERIMENTAL DETAILS

All methods were trained with AdamW and a cosine-annealed learning-rate schedule, using a batch size of 4. The learning rate was fixed at $1e-4$ for all experiments. For CRONOS, we used 10 integration steps during inference, see appendix ( B) for ablations. To ensure fair comparison, we ran each experiment three times with different validation splits and the same random seed within each split. We trained each model for the same number of epochs, selecting the checkpoint with the best NRMSE. We report NRMSE, PSNR and SSIM for each experiment. Further details can be found C.2, such as how we fix the random masking for ACDC and ISLES. Furthermore, we use Last Context Image (LCI), a heuristic that serves as strong heuristic baseline. This is simply the last available image in the sequence. This baseline is medically motivated, as it serves as a part of medical decision making when looking at longitudinal series (see Therasse et al. (2000)). Furthermore, in a setting with slowly changing anatomy, this is surprisingly strong.

### C.2    RANDOM MASKING

For ACDC and ISLES, we randomly omit context images during both training and validation, to simulate irregular sampling. Since we believe we are the first to benchmark methods in this very specific irregular setting, we **highlight a potentially grave pitfall**: If validation masks are resampled at each validation epoch, even with a fixed seed, the masking evaluation metrics change every time, which is exacerbated by our small validation set. This variability affects even the trivial LCI baseline and makes "best" epoch selection arbitrary. Since the validation set is small, context sequences can be extremely sparse or dense, causing the LCI baseline's performance to fluctuate drastically[3]. To avoid this issue, we generate one fixed set of masks per split (using a single seed) and reuse those exact masks for every model at every validation epoch. This ensures consistent validation conditions, meaningful epoch selection, and fully reproducible and interpretable validation results. In all cases, models were selected by the lowest validation $MSE$ and then evaluated on the held-out test set.

### C.3    NETWORK ARCHITECTURE

Our default network $v_\theta$ is a four-scale 3D UNet with residual blocks and FiLM time conditioning. The input $X \in \mathbb{R}^{B \times (C \times T) \times D \times H \times W}$ (typically $C=1$) stacks $T$ context volumes along channels and is linearly mixed by a $1 \times 1 \times 1$ Conv3D to $C_{\text{stem}}=32$. The encoder uses channel expansion rates $[1, 1, 2, 4]$ with one ResBlocks per scale (Conv3D($k=3$)–GN(8)–SiLU–Conv3D($k=3$)–GN(8), with a $1 \times 1 \times 1$ projection on the skip if needed); downsampling is Conv3D($k=3$, $s=2$). The bottleneck has one ResBlocks and optional windowed self-attention over flattened spatial tokens (window $8^3$, $h=4$ heads, head dim 64), followed by a final ResBlock. The decoder upsamples trilinearly by 2, applies Conv3D($k=3$), concatenates the encoder skip, and mirrors the two-ResBlock pattern with channel width equal to the reversed down blocks. FiLM modulation is injected after the first GN in every block: $\hat{h} = \alpha_\ell \odot \text{GN}(h) + \beta_\ell + h$, where $(\alpha_\ell, \beta_\ell)$ come from an MLP over a fixed-dimensional

---

[3]In natural imaging, validation sets are much larger, so random fluctuations are less severe. In medical imaging, however, smaller validation sets make these fluctuations significant.

conditioning code; either Fourier features of the scalar flow step $\tau \in [0, 1]$ for the irregular setting, or the summed Fourier encodings of real timestamps $\sum_{i=1}^{T} \gamma(t_i)/T$ for the continuous setting. All convolutions use padding to preserve spatial sizes; normalization is GroupNorm(8), activation is SiLU, and skips are concatenative with a $1{\times}1{\times}1$ channel-align conv. The head is Conv3D(32→32, $k{=}3$)–SiLU–Conv3D(32→1, $k{=}1$) producing a single-channel voxelwise velocity field.

# D ADDENDA

## D.1 USE OF LLMS

An LLM was used to help rephrase and polish this work.

