# OpenReview forum: "CRONOS: Continuous time reconstruction for 4D medical longitudinal series"
_ICLR.cc/2026/Conference — ICLR 2026 Poster_

### Official Review · Reviewer_H1U2 · 2025-10-27

**Soundness:** 3
**Presentation:** 3
**Contribution:** 2
**Rating:** 4
**Confidence:** 3

**Summary:**

This paper focuses on temporal evolution of 3d medical imaging, given traditional works are either reliant on single prior scan or fixed grid time. Also they further make prediction on voxel level rather than traditionally global level. The method used is flow matching which learns dynamics in ODE field. Experiments on three medical imaging datasets showed improvement.

**Strengths:**

- The idea of continuous sequence-to-image forecasting model for 3D medical imaging is interesting, as there were few that tried to address both discrete and continuous space.
- Demonstrates multi-context learning beyond disease-specific priors. It seems to generalize to different dataset types (cardiac, stroke, tumor).

**Weaknesses:**

- The temporal model feels suboptimal. For continuous setting they compute time conditioning as the mean of Fourier features: $\text{Enc}(t)=\frac{1}{T}\sum_{i=1}^T\gamma(t_i)$ and inject via FiLM. But averaging removes ordering and emphasizes the centroid of context times, not the relative spacing or which context frame is earlier/later. Two very different temporal configurations (e.g., times [0,1,10] vs [4,5,6]) can have similar mean embeddings but imply dramatically different dynamics. Time information relevant for rate estimation is the vector of deltas $\{t_{\text{target}}-t_i\}$, not the mean. The current encoding conflates sequences of varying lengths and spacing. So model may ignore order and rely more on spatial cues; can fail when relative timing matters (e.g., when early vs late context frames carry different predictive value).
- they stack T context volumes into $X_0=[I_1,\dots,I_T]$ and broadcast $I_{\text{target}} to X_1$. The training target in velocity space is always $I_{\text{target}}-I$ (element-wise per-channel difference), i.e., the same subtraction applied to each context channel. Different context frames correspond to different $\Delta t_i$. Using the same raw difference $I_{\text{target}}-I_i$ as a per-channel supervision makes the network learn different displacements for each channel that are unrelated to any consistent per-channel dynamics. This places the burden on $v_\theta$ to internally down-weight or rescale channels based on timestamps—again brittle and underconstrained.
- Overall one major concern would be whether continuous-time modeling actually adds value beyond what a well-tuned discrete model with adaptive interpolation can achieve. Why not simply use a dense-grid discrete model with a learned interpolation kernel

**Questions:**

the paper’s computational efficiency claims appear generous? While CRONOS uses less memory than ViViT (which OOMs), Table 8 shows it still uses 7–8 GB on a 40 GB GPU—more than ConvLSTM and comparable to SimVP

---

> ### Author Response · Authors · 2025-11-18
> **Response to Reviewer 4**
>
> We thank Reviewer 4 for the detailed and constructive comments. We address each point:
>
> W1: We agree that temporal encoding is an important direction for future work for the continuous variant, but there is a misunderstanding in the reviewer’s description. The context volumes are always provided in their chronological order, so the model does not treat them as an unordered set. Moreover, the fourier features are calculated per timestamp, and these high frequency embeddings differ substantially between configurations such as [0,1,10] and [4,5,6]. The mean aggregation is used only on the high-dimensional fourier features to obtain a global conditioning signal, similar in spirit to Neural Processes or deep sets, which was chosen for simplicity. Additionally, the number of times is smaller than the feature size of the U-Net.
>  Future work may include additional ways to condition the network on the time.
>
> W2: The supervision is not shared across channels. Each context frame has its own velocity component in the output tensor $v_\theta \in \mathbb{R}^{T  \times S}$ (with $T$ the amount of frames, and $S$ the spatial dims), so the model learns $T$ distinct displacements, we clarified the velocity in the paper. This design is intentional, as it allows the model to process all context frames jointly, while still predicting per-frame velocities. Alternative aggregation strategies are possible (see Table 4). Additional schemes are possible future extensions.
>
> W3: Our results show that the continuous variant provides clear benefits in the continuous setting: in Table 3 (and the appendix), it consistently outperforms the discrete variant.
> On the discrete setting, both variants perform equally well (Table 2), so the discrete variant remains appropriate there.
> Additionally, the continuous variant is computationally more efficient in training and inference time, as well as in memory usage.
> When the grid embedding forces large grid ranges (Eqs. 12-18), the continuous variant scales more favourably than the discrete variant.
> The continuous variant is therefore not a replacement, but a complementary part of the framework.
>
> Q1: We have clarified the performance claims in the paper. CRONOS is roughly on par with spatio-temporal (ST) methods \textit{during training} in terms of max compute (Table 8), but scale better with the number of grid points than the other methods (Figure 1b). Also, we added the memory requirements for smaller feature sizes in the appendix (Table 5), where the performance of CRONOS only drops slightly with smaller feature size, but the memory footprint falls below the ST baselines.

---

### Official Review · Reviewer_EAfG · 2025-10-30

**Soundness:** 3
**Presentation:** 3
**Contribution:** 2
**Rating:** 4
**Confidence:** 4

**Summary:**

This paper tackles the task of forecasting a 3D medical image at a future time given multiple prior 3D scans (a longitudinal series), even when the input scans have irregular timing. The authors propose CRONOS, a unified many-to-one framework that models continuous time by learning a spatio-temporal velocity field to warp all input volumes toward the target volume. This approach adapts the Flow Matching paradigm to operate directly in voxel space, enabling real-valued timestamp inputs within a single model. Experiments on three public datasets (cine MRI, perfusion CT, and longitudinal brain MRI) show that CRONOS outperforms several baseline methods in image quality metrics while using significantly less memory than diffusion-based approaches. The authors highlight that this is the first method to support multi-context continuous-time forecasting for 3D medical images, and they promise to release code and evaluation protocols for reproducibility.

**Strengths:**

Addresses an important, under-explored problem: The paper targets 3D medical image sequence forecasting from multiple prior scans at irregular time intervals – a challenging and clinically relevant task (e.g. disease progression) that has received limited attention.

Unified continuous-time approach: CRONOS is a novel framework combining multiple-context inputs with continuous-time prediction in one model. It appears to be the first method to handle real-valued time stamps and multiple input volumes together in 3D image forecasting, filling a gap in the literature.

Efficient flow-based method: The use of Flow Matching (ODE-based velocity learning) instead of iterative diffusion leads to a simpler, memory-efficient solution. CRONOS scales to long input sequences with substantially lower GPU memory usage and runtime, without needing expensive time-step sampling.

Strong results on diverse datasets: The proposed model demonstrates improved performance over prior baselines (including deep video models and a strong last-scan heuristic) across three different modalities. It consistently achieves higher image similarity metrics (e.g. SSIM) on cine MRI, CT perfusion, and MRI longitudinal datasets, indicating robustness across varied medical imaging scenarios.

Thorough evaluation and reproducibility: The authors conduct experiments on multiple public datasets and compare against several baseline approaches. They also commit to releasing code and standardized evaluation protocols, which enhances the work’s credibility and potential impact as a benchmark for multi-context forecasting.

**Weaknesses:**

Limited algorithmic novelty (major): The approach offers minimal new methodology, essentially applying the existing Flow Matching paradigm to the many-to-one 3D forecasting setting without introducing new algorithmic contributions. The core idea (learning a velocity field via flow matching) is borrowed from prior work, with novelty lying primarily in its application to this domain rather than in the technique itself.

Missing baseline comparisons (major): The evaluation lacks comparisons with several relevant continuous-time or generative baseline methods. Notably, recent (core-level related) approaches like STDiff (Ye & Bilodeau, 2023), LoCI-DiffCom (Zhu et al., 2024), and ImageFlowNet (Liu et al., 2025) are not included, leaving it unclear how CRONOS performs relative to the state-of-the-art in continuous video/volume forecasting.

Unjustified design choice: CRONOS uses a single shared velocity field to warp all input contexts toward the target, but the paper provides no clear justification or ablation for this design. It is not explained why one global velocity field (applied to all past scans) is sufficient, as opposed to using separate or time-dependent velocity fields. This raises concerns that the model might be overly restrictive or suboptimal in capturing differing motions from each context scan.

Continuous-time generalization unproven: While the method is designed for continuous time, the experiments do not explicitly validate interpolation or extrapolation to truly arbitrary time points. There is no demonstration of predicting intermediate frames between given scans or forecasting beyond the maximum training interval. Thus, the claimed continuous-time capability remains speculative, with no evidence that the model can generalize to time points outside the training distribution.

Deterministic outputs (no uncertainty modeling): The framework produces a single deterministic prediction for a given input set and time, with no probabilistic modeling or uncertainty quantification. Unlike stochastic generative models (e.g. diffusion-based methods), CRONOS cannot express multiple plausible future outcomes or estimate confidence in its predictions. This lack of uncertainty modeling is a drawback for medical forecasting, where understanding predictive uncertainty is often important.

Generality claims not well-supported: The authors suggest the approach is modality-agnostic and broadly applicable, but evidence is limited to the medical imaging domain. All experiments are on medical scans; there is no validation on other data (e.g. natural video or other 4D datasets) to support the claim of general applicability. The strong generalization claims feel somewhat overstated without testing beyond the specific modalities presented.

No interpretability analysis of flows: The learned velocity fields (which are central to the method) are not analyzed or visualized in the paper. There is no examination of whether the predicted flow patterns are physically or biologically plausible. The absence of any qualitative assessment of these flows is a missed opportunity – it remains unclear if CRONOS learns meaningful motion patterns or if the velocity field could provide insights (e.g. highlighting anatomical changes or disease progression).

**Questions:**

Novelty: What specific technical contributions does CRONOS introduce beyond applying standard Flow Matching to this multi-volume forecasting task? (For example, are there any new model components or training strategies, or is the contribution primarily the problem setup?)

Baselines: Have the authors evaluated CRONOS against recent continuous-time or generative baselines such as STDiff (Ye & Bilodeau, 2023), LoCI-DiffCom (Zhu et al., 2024), ImageFlowNet (Liu et al., 2025), or more state-of-the-art studies? If not, what was the rationale for omitting these comparisons, and how would the authors expect CRONOS to perform relative to those methods?

Shared velocity field: Why was a single shared velocity field used to warp all input volumes toward the target? Did the authors consider using distinct velocity fields per context or a time-varying velocity field, and if so, how might that affect the results?

Continuous-time evaluation: Can CRONOS generate predictions at arbitrary time points outside the training timestamps (for example, intermediate time interpolation or longer-term extrapolation)? If this was not tested, could the authors clarify how they expect the model to behave for times beyond the training range?

Uncertainty modeling: Do the authors plan to incorporate any form of uncertainty or stochasticity into CRONOS? For instance, could a probabilistic extension (e.g. a diffusion-based, VAE, or flow model) be used to capture multiple plausible future outcomes and quantify prediction confidence?

Generality beyond medical domain: Has CRONOS been evaluated on non-medical temporal imaging data (such as natural videos or other 4D sequences) to demonstrate its generality beyond the medical domain? If not, would the authors consider tempering the claims of broad applicability until such evidence is provided, or could they outline how the method might generalize to other domains?

Flow field interpretability: Have the authors analyzed or visualized the learned velocity fields to assess their interpretability or biological plausibility? If not, could providing such an analysis (e.g. showing whether the predicted flow aligns with known anatomical motion or expected deformation patterns) strengthen the paper’s conclusions?

---

> ### Author Response · Authors · 2025-11-18
> **Response to Reviewer 3**
>
> We thank the reviewer for the constructive and thorough questions.
> Below we clarify each point and address all questions and weaknesses. Where appropriate, we group related responses:
>
> Q/W1 novelty: CRONOS builds on Flow Matching, but its technical novelty lies in adapting FM to longitudinal 3D forecasting with multi-context conditioning, irregular sampling and missingness. In the discrete formulation, we introduce a concrete setup for sequence-to-image forecasting: context times are embedded onto a finite grid, missing frames are handled via last-observed-carry-forward, and we supervise per-frame velocities, jointly across the entire sequence.
> In the continuous variant, we extend FM to real-valued timestamps, where the classical formulation has a single, non-physical $\tau \in [0,1]$, where we propose to use $\mathcal{T}_\tau$, a vector of times. We therefore propose a way to condition on multiple images at arbitrary timepoints.
>
> Q/W2 (baselines):  The rationale for omitting these methods is explained in Table 1, due to technical or due to incompatibility reasons. Preliminary experiments, as well as recent evidence (MICCAI paper #1363 2025) support the decision to use flow matching. We attribute the improved performance due to the inherent properties of medical imaging, where changes are small. ImageFlowNet in particular, while interesting, may have difficulties scaling as it uses Neural ODEs in image space. An adaptation of ImageFlowNet to Flow Matching would likely reach at least the performance of the LCI baseline, but it would still be limited to single‐image conditioning and therefore unable to leverage multiple context frames.
>
> Q/W3 (velocity field): The model does not use a single velocity field, but a time-dependent velocity field $v_\theta \in \mathbb{R}^{T\times S}$, we clarified that in the paper. The network therefore learns a frame-specific displacement while  conditioning jointly on the entire sequence. Both the discrete and the continuous versions have therefore a time-dependent velocity field.
>
> Q/W4 (extrapolation): CRONOS can be queried for arbitrary timestamps, but meaningful extrapolation depends entirely on what the model was trained to observe. In our ACDC setup, the sequences contain only the ED (end-dialostic) $\to$ ES (end-systolic), so the model never sees a full periodic cardiac cycle and receives no supervision beyond the target frame. When evaluated beyond that range, predictions converge to the last predicted state with no further progression. This likely reflects the training regime rather than a limitation of the method. It also may be dataset specific (e.g. in ISLES, intensities stabilize and the long horizon output remains similar). Extending the evaluations to more diverse settings is part of future work.
>
> Q/W5 (stochasticity): CRONOS already supports stochastic training and generation. During training, we can inject noise into the velocity supervision (see ablation Table 7 for details for stochastic training), and during inference we can replace the deterministic ODE solver with a stochastic variant such as Euler-Maruyama (i.e. SDE in Algorithms 1 and 2). We find that CRONOS is robust under moderate noise. But our evaluation focuses on deterministic forecasts to remain comparable with prior work, and because these datasets only provide a single future trajectory. Therefore this is a limitation of the evaluation setting, and in future work we will test CRONOS on stochastic forecasting.
>
> Q/W6 (generalization beyond medical): Our work primarily focuses on medical longitudinal imaging. Within that, we have shown generalizability across modalities and datasets. We specified wherever applicable, the wording that our claims focus on medical imaging. However, as there is no medical prior in our method, CRONOS may be able to generalize to settings where changes in the data are small.  Evaluating CRONOS  outside the medical domain would strengthen the robustness and interpretability. We would welcome any suitable benchmark suggestions for future work.
>
> Q/W7 (flow interpretation): We have now included a flow-field visualization in Figure 5. The strongest velocity magnitudes appear around the myocardium, matching regions of expected physiological change. These areas also coincide with the main residual difference between the target and LCI. This qualitative check is informative, but it does not change our conclusions: as CRONOS already outperforms the LCI quantitatively, it must capture meaningful temporal changes.

---

### Official Review · Reviewer_Sqka · 2025-10-31

**Soundness:** 3
**Presentation:** 3
**Contribution:** 3
**Rating:** 6
**Confidence:** 3

**Summary:**

The paper introduces CRONOS which adapts Flow Matching to medical imaging. The main contributions are: (i) temporal broadcasting which treats multiple context scans as the source distribution and the broadcast target scan as the destination, (ii) dual formulation giving both discrete and continuous variants, and a (iii) unified framework handling multiple modalities without disease-specific assumptions.
Overall, the paper treats the problem of predicting future 3D medical scans from multiple past observations at irregular time intervals. This is a clinically relevant problem. Most existing methods are limited to single input images and lack continuous-time predictions.

**Strengths:**

1. Problem Relevance: The paper addresses a critical problem in 3D medical imaging by proposing a unified framework for continuous sequence-to-image forecasting.

2. Technical contribution: the method makes use of Flow Matching applied to the medical imaging domain through temporal broadcasting. The theoritical formulation is clear and well defined.

3. Performance. CRONOS shows strong performance over LCI baseline and the spatio-temporal baselines across all three datasets.

3. Evaluation: The paper proposes a comprehensive evaluation on three different datasets (ACDC, ISLES, LUMIERE) with different modalities (Cine-MRI, perfusion CT, longitudinal MRI). These experiments show good generalization capabilities of CRONOS.

**Weaknesses:**

1. Lack of expert validation: only voxel-wise metrics (NRMSE, PSNR, SSIM) are provided to assess performance but no clinical expert validation is shown nor other relevant anatomical metrics.

2. Limited improvement: CRONOS scores 94.51% vs. LCI's 92.79% SSIM on ACDC, which appears to be a small gain given the added computational cost.

3. Methodological issues: there are concerns the LOCF (Last Observed Carry Forward) filling strategy (eq 19) may introduce artifacts. Even though it is demonstrated as a key component in the ablation studies, no comparison with other approaches is shown.

4. Limited discussion of failures: The paper does not provide insights into when the model might fail.

**Questions:**

1. Continuous vs discrete trade-offs: there appears to be inconsistencies in Table 3 and Table 2 where continuous CRONOS (Table 3) outperforms discrete version on the continuous ACDC ablation, but not necessarily on Table 2. This raises the question when is best to use continuous vs discrete alternative?

2. Could you show what the performance means in terms of clinical impact ?

3. How does the performance scales with the number of frames and spatial resolution ?

---

> ### Author Response · Authors · 2025-11-18
> **Response to Reviewer 2**
>
> We thank Reviewer 2 for the constructive feedback. Below we address the questions and provide clarifications, as well as additional results where appropriate.
>
> W1: We follow prior work in spatio-temporal medical image prediction and report voxel-wise metrics, which remain the established and comparable evaluation measures in this setting. Anatomical metrics and clinical reader studies are highly task-dependent and difficult to standardize  across the three modalities we evaluated. Such assessments require dedicated clinical protocols and are outside of the scope of this technical contribution, but would be appropriate for future clinical studies.
>
> W2: LCI is the last image carried forward, and is not a prediction.
> Even modest improvements over LCI are therefore meaningful, as they indicate the model is capturing non-trivial progression rather than simply copying the last frame.
>
> W3: We have added an ablation comparing LOCF  to a zero fill+mask variant. The masked formulation trains substantially less stably and takes longer to converge, and within this setting, it performs slightly worse. Future work could further consider filling strategies for the discrete CRONOS. Furthermore, the continuous CRONOS variant makes the LOCF obsolete.
>
> Q1: The flexibility of our formulation allows both discrete and continuous conditioning, and the choice depends on the temporal characteristics of the dataset. For discrete sampling (Table 2) both variants perform similarly, but the continuous variant is more efficient in terms of max memory and training time. When the sampling becomes highly irregular (Table 3), the discrete variant is constrained, in contrast to the continuous version. Further analysis of differences is part of future work. Also, the continuous variant does not need the LOCF.
>
> Q2: Assessing direct clinical impact requires dedicated reader studies and outcome-driven evals, which beyond the scope of this work. We view CRONOS as an enabling step, and full clinical evaluation is a natural direction for future work.
>
> Q3: We added the performance increase in terms of missing frames, from the front and the back (Figure 6).

---

### Official Review · Reviewer_YVaG · 2025-11-01

**Soundness:** 3
**Presentation:** 3
**Contribution:** 3
**Rating:** 6
**Confidence:** 5

**Summary:**

This paper introduces CRONOS, a flow-based framework for 3D medical image forecasting that learns continuous spatio-temporal dynamics from longitudinal scans.Unlike prior models restricted to single-context or grid-aligned timepoints, CRONOS handles both discrete and continuous timestamps by learning a spatio-temporal velocity field that transports a stack of past volumes toward a future target scan. Technically, the authors reinterpret Flow Matching (FM) (Lipman et al., 2023) as a sequence-to-image transport problem:
$X_0 = [I_1, …, I_T],\qquad X_1 = [I_{\text{target}}, …, I_{\text{target}}],$ and train a 3D U-Net $v_\theta(X_\tau,T_\tau)$ to approximate voxel-space velocities between context and target. The authors suggested two complementary variants:
1. Discrete CRONOS: uses grid embedding + last-observation-carry-forward (LOCF) to fill missing frames.
2. Continuous CRONOS: conditions directly on real timestamps via Fourier embeddings and FiLM.

Experiments on three public 3D datasets—ACDC (cardiac MRI), ISLES (perfusion CT), and LUMIERE (longitudinal glioma MRI)—show consistent improvements over ConvLSTM, SimVP, ViViT, NODE-LSTM, and the simple Last Context Image (LCI) baseline. CRONOS maintains competitive compute cost, yields sharper reconstructions, and performs best when irregular timestamps are available.

**Strengths:**

1. Longitudinal 4D imaging is central to disease progression analysis, yet remains underexplored by the ML community.
2. Extending FM from noise-to-sample to multi-context-to-target volumetric flows is conceptually creative and unifies discrete and continuous-time modeling within one ODE framework.
3. The discrete/continuous variants and Fourier-time conditioning offer a simple yet flexible approach adaptable to both regularly and irregularly sampled medical series.

**Weaknesses:**

1. The “velocity loss” $\|v_\theta(X_\tau,T_\tau)-(I_{\text{target}}-I)\|^2$ treats voxel-intensity differences as flow supervision.
This deviates from FM’s probabilistic transport interpretation and lacks theoretical grounding in physical or latent-space dynamics.
2. ConvLSTM, SimVP, and ViViT are optimized for long, dense 2D videos, not short 3D series. Their poor results may reflect mismatch rather than true inferiority.
3. The “continuous” setting is created by subsampling ACDC; no genuinely irregular clinical series are used. Gains may arise from richer embeddings rather than true continuous modeling.

**Questions:**

1. Does the learned velocity field correspond to interpretable motion (can it be visualized or regularized)?
2. Could CRONOS handle autoregressive multi-step prediction, not just one-step forecasting?
3. Have you compared with a “zero + mask” imputation instead of LOCF to ensure fairness?

---

> ### Author Response · Authors · 2025-11-18
> **Response to Reviewer 1**
>
> We thank the reviewer for the constructive and helpful comments.
>
> W1: The proposed loss is the standard FM loss applied in voxel-space, and follows the transport formulation of FM without modification. CRONOS models intensity changes directly, which is especially appropriate for datasets such as ISLES and LUMIERE.
> We clarified that CRONOS supports stochastic training as well (see Table 7 for ablations), which is consistent with the original formulation. Our modeling choice in image space is motivated by the fact that latent encodings may distort fine-scale structures, even in areas where changes are not present. A more in-depth theoretical analysis is an interesting direction for future work.
>
> W2: The three methods are not inherently mismatched to medical data. Both ViViT and ConvLSTMs have been applied to prior works in spatio-temporal medical imaging. Solely SimVP has been newly applied to medical 4D data in this work. In our view, this highlights the need for a stable, scalable  and simple method tailored to medical spatio-temporal forecasting.
>
> W3: The continuous setting is based on controlled subsampling of ACDC, which allows us to isolate the effects on temporal irregularity. We agree that clinical series can exhibit stronger irregularity; however, most readily available ST datasets can still be embedded to a common temporal grid (Eqs 12-18, Appendix), albeit at the cost of larger grid sizes.
>
> Q1: Yes, we can visualize these velocities, and we added an example in Figure 5, where this corresponds to physical motion. Regularization is possible (e.g. we have noise regularization in Table 7) , but a principled analysis would be a natural direction for future work.
>
> Q2:  CRONOS can be applied autoregressively: once a volume is predicted, it can be appended to the previous context, and the first context frame dismissed. In preliminary tests, this works as expected, but accumulates errors and increases computation time linearly with the prediction horizon. Whether multi-step forecasting has benefits would be suited for future extensions.
>
> Q3: Yes, we added this to Table 4. We find that the masking strategy converges more slowly and is less stable in training, and achieves slightly worse performance. Assessing different filling strategies would be part of future work for the discrete variant.

---

### Author Response · Authors · 2025-11-27
**Summary Comment**

We thank the Reviewers again for their thorough and constructive feedback. We have replied to each comment thoroughly and summarized them here for the AC to assist their evaluation in these difficult times.

Across the reviews, there is a clear consensus from all reviewers on the strengths of our submission:

1. The task is considered central (for disease progression) and remains underexplored (R1,R2,R3,R4).
2. Reviewers highlight the technical contribution and creative methodology (R1,R2,R3),
3. Also noted was the efficiency and performance of our approach (R2,R3),
4. We were also lauded for our thorough evaluation with generalization across multiple medical datasets (R2,R3,R4).

Together, these points indicate that the proposed framework forms a strong and reliable baseline for this underexplored setting.

We addressed all weaknesses and questions raised. Several were overlapping, including the precise form of velocity supervision, the interpretation and visualization of the velocity field (where we added additional qualitative Figures), and the role of stochasticity versus determinism in our framework, which we all clarified in the manuscript. For details of the responses to all the concerns, we refer to the respective responses in the discussion.

We hope this summary, together with our clarifications, assists the AC in their assessment. We thank you once again for your efforts during this challenging review process.

---

### Meta-Review · Area_Chair_JW96 · 2025-12-17

**Summary:**

Reviewers agreed that CRONOS tackles a clinically relevant and underexplored problem and found the flow-matching formulation effective across multiple datasets. Minor concerns remained about limited methodological novelty beyond adapting existing flow-matching ideas, incomplete baseline coverage against recent continuous-time/generative models, and insufficient evidence for true continuous-time generalization on genuinely irregular clinical data. Overall, this is a paper with borderline scores. The area chair thinks the work is complete and solid, and it has merit of being accepted at ICLR.

**Reviewer Concerns:**

The rebuttal addressed several technical concerns by clarifying the formulation and supervision of the spatio-temporal velocity field, adding qualitative visualizations of learned flows, and explaining the design choices behind discrete versus continuous variants and the use of LOCF versus masking. It also clarified that the model supports time-dependent velocities, stochastic training variants, and autoregressive rollouts, and provided additional ablations to support these points.

Minor concerns remain around higher-level issues. Some reviewers were not fully convinced of the methodological novelty beyond adapting flow matching to a new setting, questioned the absence of several recent continuous-time or generative baselines, and felt that the evidence for true continuous-time generalization (beyond subsampled grids) was limited. In addition, the lack of clinical or anatomy-aware evaluation and a more explicit discussion of failure cases and practical impact were not fully resolved.

**Reviewer Scores:**

Reviewers agreed that CRONOS addresses an important and underexplored problem and found the flow-matching formulation technically sound and effective across multiple datasets.

The rebuttal clarified several methodological points, including velocity-field interpretation, LOCF versus masking, the time-dependent nature of the velocity field, and the roles of discrete versus continuous variants, and added qualitative visualizations and ablations.

Some high-level concerns remain only partially resolved: some reviewers viewed the contribution as primarily an adaptation of existing flow-matching ideas, and noted limited clinical/anatomical validation and failure-mode analysis.

---

### Decision · Program_Chairs · 2026-01-26

Accept (Poster)